# Characterisation of class VI TRIM RING domains: linking RING activity to C-terminal domain identity

Rebecca V Stevens, Diego Esposito, Katrin Rittinger

**TRIM E3 ubiquitin ligases regulate multiple cellular processes, and their dysfunction is linked to disease. They are characterised by a conserved N-terminal tripartite motif comprising a RING, B-box domains, and a coiled-coil region, with C-terminal domains often mediating substrate recruitment. TRIM proteins are grouped into 11 classes based on C-terminal domain identity. Class VI TRIMs, TRIM24, TRIM33, and TRIM28, have been described as transcriptional regulators, a function linked to their C-terminal plant homeodomain and bromodomain, and independent of their ubiquitination activity. It is unclear whether E3 ligase activity is regulated in family members where the C-terminal domains function independently. Here, we provide a detailed biochemical characterisation of the RING domains of class VI TRIMs and describe the solution structure of the TRIM28 RING. Our study reveals a lack of activity of the isolated RING domains, which may be linked to the absence of self-association. We propose that class VI TRIMs exist in an inactive state and require additional regulatory events to stimulate E3 ligase activity, ensuring that associated chromatin-remodelling factors are not injudiciously degraded.**

## Introduction

Modification of proteins with either a single ubiquitin molecule or poly-ubiquitin chains is a regulatory signalling mechanism integral to many cellular functions. Ubiquitin becomes covalently attached to a substrate lysine residue in a process catalysed by a multienzyme cascade, involving E1 activating enzymes, E2 conjugating enzymes, and E3 ligase enzymes. In humans, there are 2 E1 activating enzymes, 35 E2 conjugating enzymes, and more than 600 E3 ligase enzymes. Ubiquitin ligases provide substrate specificity and are divided into three families, based on the mechanism used to transfer ubiquitin onto the substrate (1). The largest family of E3s, RING-type E3 ligases, act as adaptors and catalyse the direct transfer of ubiquitin from the E2–Ub conjugate to the substrate (2, 3).

The TRIM protein family constitutes the largest subfamily of RING E3 ligases, with more than 70 family members in humans that regulate many cellular processes. TRIM proteins are characterised by a conserved N-terminal tripartite motif and a variable C-terminal region and can be subcategorised into 11 classes based on C-terminal domain identity (4, 5, 6). The tripartite motif (also called RBCC) is composed of a RING domain, one or two B-box domains, and a coiled-coil region (7). The RING domain is a specialised zinc finger and is the minimal functional unit required to mediate ubiquitin transfer. B-box domains are zinc-binding motifs, which like the RING domain coordinate two zinc ions. The precise function of B-box domains is not fully understood, but they are often considered to mediate protein–protein interactions; in this context, they are involved in autoinhibition of TRIM21 (8) and homo-oligomerisation of TRIM5α (9). The coiled-coil domain drives homo-dimerisation, which occurs in an antiparallel fashion (10, 11) and may also facilitate hetero-oligomerisation of TRIM proteins (12).

The C-terminal domains of TRIM proteins are assumed to facilitate substrate recruitment, as has been shown for the PRYSPRY domain, which describes the most populous class of TRIMs. However, at least one class of TRIM proteins has an additional, distinct cellular role derived from its C-terminal domains. The class VI TRIM proteins comprising TRIM24 (TIF1α), TRIM28 (TIF1β or KAP1), and TRIM33 (TIF1γ) function as chromatin-associated transcriptional co-regulators via their C-terminal domains, a plant homeodomain (PHD) and a bromodomain (13, 14, 15, 16). The PHD is a zinc finger domain, which in some cases, possesses SUMO E3 ligase activity (17, 18), whereas canonical bromodomains recognise acetylated lysine residues on the N-terminal tails of histone proteins (19, 20, 21, 22, 23). TRIM28, the most extensively studied member of this family, is recruited to promoters by KRAB domain-containing transcription factors, via interaction with the RBCC (24). The sumoylation of the bromodomain, by the adjacent PHD domain, facilitates the recruitment of the Nucleosome Remodelling Deacetylase and the H3-K9 histone methyltransferase, SETDB1 (25). This chromatin-remodelling complex functions to establish a repressive chromatin state. In addition, multiple studies have highlighted an important role of TRIM28 in the repression of endogenous viruses (26, 27, 28, 29).

Although studies of class VI TRIM proteins to date have primarily focussed on their roles as transcriptional regulators, they have also been described to function as ubiquitin E3 ligases. TRIM24 and TRIM28

Molecular Structure of Cell Signalling Laboratory, The Francis Crick Institute, London, UK

Correspondence: katrin.rittinger@crick.ac.uk

have both been reported as part of a network of E3 ligases, which regulate the stability of the tumour suppressor protein, p53, via K48-linked ubiquitination and proteasomal degradation (30, 31, 32). TRIM33 targets nuclear β-catenin, aberrantly activated in a number of human cancers, for degradation and has been linked to ubiquitination of SMAD4 to prevent its association with SMAD2/3 (33, 34, 35).

In addition, TRIM28 has been identified as a binding partner of class I MAGE proteins, which are able to exploit the E3 ligase activity of TRIM28 in the progression of a variety of cancers (36, 37). MAGE protein binding to the coiled-coil domain of TRIM28 has been reported to prime TRIM28 for ubiquitination of p53 and AMPK, inducing their proteasomal degradation. The molecular mechanism of this activity is unclear at present, but it has been suggested that MAGE protein binding may release an autoinhibitory conformation of TRIM28, enhance catalytic activity, or act as a substrate adaptor (38).

TRIM proteins derive their E3 ligase activity from the N-terminal RING domain, and most TRIM proteins studied thus far require RING dimerisation for catalytic activity (Fig 1) (39, 40, 41). RING dimerisation of TRIMs generally requires two short helices, N- and C-terminal to the core RING domain; removal of these helices from the TRIM32 RING resulted in a monomeric inactive RING domain (39). Crystal structures of a number of TRIM RING domains indicate that these helical elements form a four-helix bundle in the dimeric RING structure. Formation of a RING dimer facilitates additional contacts with ubiquitin during E2–Ub conjugate recognition: E2 binds to the proximal RING while ubiquitin contacts both the proximal and the distal RING. These additional contacts with ubiquitin promote the formation of a "closed" conformation of the E2–Ub conjugate, priming the active site for ubiquitin transfer (42, 43, 44, 45).

Although a few individual TRIM proteins have been extensively characterised, there has been no systematic investigation of a specific subclass of TRIM proteins. It is, therefore, unclear whether there is a relationship between C-terminal domain identity and RING domain activity. Class VI TRIM proteins stand out in particular as their C-terminal domains mediate association with chromatin-remodelling complexes that do not appear to become ubiquitinated. This suggests that the E3 ligase activity of this TRIM class is tightly regulated, by an unknown mechanism, to prevent indiscriminate ubiquitination of chromatin-remodelling complexes.

To test this hypothesis, we have carried out a detailed biochemical characterisation of the RING domains of class VI TRIMs and have solved the solution structure of the TRIM28 RING. Our study shows that members of this family are unable to self-associate and do not harbour ubiquitin ligase activity on their own. Based on our data, we propose that class VI TRIM ligases may constitute inactive components of heterodimeric E3 ligase complexes or may require additional binding partners to ensure that catalytic activity is only unleashed in response to the correct stimulus.

## Results

### Activity of class VI TRIM RING domains

To assess the basic E3 ubiquitin ligase activity of class VI TRIM proteins, we produced the RING domains of TRIM24, TRIM33, and TRIM28 in isolation to rule out any autoinhibitory effects from adjacent domains present in the full-length proteins. However,

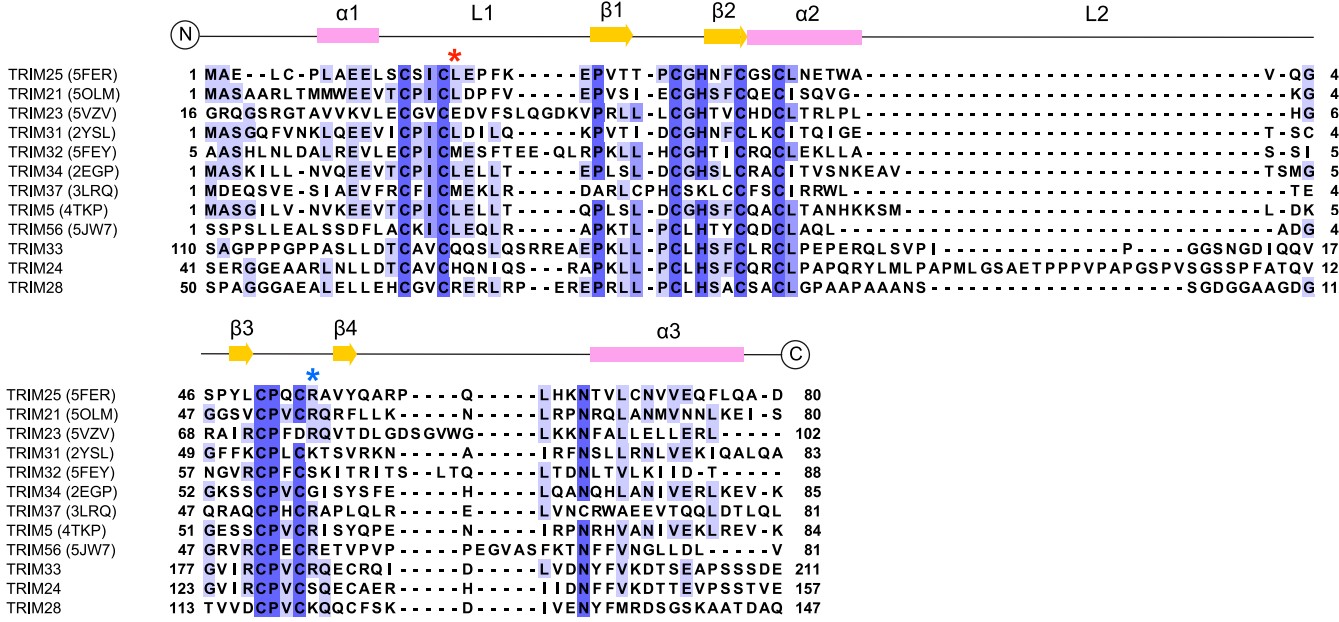

**Figure 1. Multiple sequence alignment of TRIM RING domains.**
Multiple sequence alignment of the class VI RING domains with those TRIM RING domains for which there are structures in the PDB. PDB codes are indicated in brackets. The secondary structure of the TRIM25 RING is shown above the alignment. The red asterisk denotes the conserved hydrophobic residue, which is important for E2 binding in various RINGs. The blue asterisk denotes the conserved "linchpin" residue, which forms hydrogen bonds with both E2 and ubiquitin.

given that RING domains are small and contain few lysine residues, they may be unable to perform auto-ubiquitination. Activity was, therefore, assayed using lysine-discharge assays, which monitor the ability of a given construct to activate a pre-charged E2–Ub conjugate for transfer of ubiquitin onto free lysine. We tested the activity with three different E2 enzymes: UBE2D (UbcH5), UBE2E2 (UbcH8), and UBE2E1 (UbcH6). TRIM24 has been reported to catalyse p53 ubiquitination in conjunction with UBE2E2 (30), whereas TRIM28 activity has been linked with UBE2D (36), and TRIM33 has been proposed to function with both UBE2E1 and UBE2D (34, 46). The TRIM25 RING domain was used as a positive control for UBE2D, and the TRIM32 RBCC construct for both UBE2E1 and UBE2E2.

Surprisingly, none of the class VI RINGs were able to enhance ubiquitin discharge from UBE2D–Ub above the level of the negative control. In comparison, the TRIM25 RING significantly enhanced discharge of ubiquitin from UBE2D, with almost all ubiquitin transferred onto free lysine after 30 min (Fig 2A and B). The class VI RINGs were similarly unable to enhance ubiquitin discharge from either the UBE2E1–Ub or UBE2E2–Ub conjugate, whereas the TRIM32 positive control efficiently catalysed discharge from both E2–Ub intermediates (Fig 2C–F). Increasing the concentration of the RING domains had no effect on discharge efficiency (Fig S1).

Taken together, these data indicate that the isolated class VI RINGs are unable to catalyse ubiquitin transfer in conjunction with the E2s tested.

## Oligomeric state of class VI RING domains and catalytic activity

As most TRIM RING domains require RING dimerisation for activity, we next investigated the oligomeric state of the class VI RING domains to determine whether their observed lack of activity could be linked to their oligomeric state. RING domains were analysed by analytical size-exclusion chromatography coupled with multiangle laser light scattering (SEC-MALLS). Samples were run over a range of concentrations providing light-scattering profiles consistent with a single molar mass for all class VI RING domains, indicating that these proteins were in a monomeric state at all concentrations tested (Fig 3A–C).

Although all class VI RING domains appeared monomeric by SEC-MALLS, this does not preclude dimer formation under physiological conditions. For example, the TRIM25 RING behaves as a monomer on SEC-MALLS, but dimer formation is stabilised by interaction with the charged E2–Ub conjugate. Accordingly, formation of a covalent TRIM25 tandem RING construct significantly increased the activity of this protein (39). To test whether dimerisation of the class VI RING domains could be stabilised in the same manner, a fused RING dimer of TRIM28 (TRIM28 RING–RING), the most extensively studied member of this family, was produced. Activity of the TRIM28 RING–RING was again evaluated by lysine-discharge assay but showed no increase in activity compared with the monomeric TRIM28 RING with all E2s tested (Figs 3D and E, and S2). Furthermore, comparison of the amide regions of the 1D $^1$H-NMR spectra of the isolated RING and the RING fusion construct reveals a conserved pattern of resonance peaks with similar linewidths, indicating that the individual RING domains in the RING–RING construct do not self-associate (Fig S3A).

Secondary structure prediction of the TRIM28 RING and adjacent regions indicated that although there is some helical propensity for the sequence N-terminal to the TRIM28 RING, it is unlikely that a helix would form C-terminal to the RING. To test whether this potential absence of secondary structure is responsible for the lack of dimerisation of the TRIM28 RING, even at high local concentrations (i.e., in the fused RING dimer), we created a chimera in which the core RING domain of TRIM28 was inserted between the helices of the TRIM32 and TRIM2 RING domains, which are both constitutively dimeric. However, both chimeras were insoluble, suggesting that the helical RING dimerisation modules are not easily transferable between RINGs.

All class VI RING domains are predicted to contain a relatively long unstructured loop (L2) between the core α-helix (α2) and the second zinc coordination site (Fig 1), which is not present in other structurally characterised active RINGs such as TRIM32, TRIM25, or TRIM5α. We speculated that this loop might fold back to occlude the E2-binding site. To test this model, a TRIM28 RING mutant was produced where most of this loop was removed (TRIM28 RINGΔL2) and tested for activity using lysine-discharge assays. However, no increase in the rate of ubiquitin discharge was observed for this mutant compared with the wild-type protein with all E2s tested (Figs 3D and E, and S2).

Finally, to exclude the possibility that regions outside the RING are required for catalytic activity, we assayed full-length TRIM28 in the apo form and bound to the MAGE homology domain (MHD) of MAGE-C2. It neither showed any increase in activity over the isolated RING as judged by lysine-discharge assays (Fig 3F and G). Similarly, auto-ubiquitination assays using full-length TRIM28 and a wide range of E2s did not provide any evidence for catalytic activity (Fig S4).

Taken together, these data show that none of the class VI RING domains have any propensity to homo-dimerise, not even when covalently fused to one another. Furthermore, our experiments suggest that the extended L2 loop region that is present in class VI TRIM proteins (Fig 1) does not affect the activity through intramolecular autoinhibitory interactions. Similarly, regions outside the RING domain do not appear to contribute to catalytic activity, not even in the presence of MAGE proteins, which have been described to modulate the activity of TRIM28, indicating that the observed lack of activity is an intrinsic property of the RING domains.

## The structure of the TRIM28 RING domain

To determine whether there are identifiable structural features that deviate from the canonical RING fold, which may explain the impaired activity of class VI TRIM proteins, we characterised the TRIM28 RING domain by NMR spectroscopy. Initial analysis of the 2D $^1$H-$^{15}$N HSQC spectrum reveals a dispersed pattern of cross-peaks typical of a folded protein, with a linewidth compatible with the monomeric state of TRIM28 observed by SEC-MALLS (Fig S5A).

Full resonance assignment of the TRIM28 RING, using well-established triple resonance NMR methods, was carried out on a $^{13}$C, $^{15}$N–labelled sample. Complete chemical shift assignment was obtained for residues 54–145, including Asn $^{15}$Nδ and Gln $^{15}$Nε side-chain resonances; residues remaining from GST-tag cleavage (Gly52-Pro53) were not assigned. All cysteine side-chain Cα

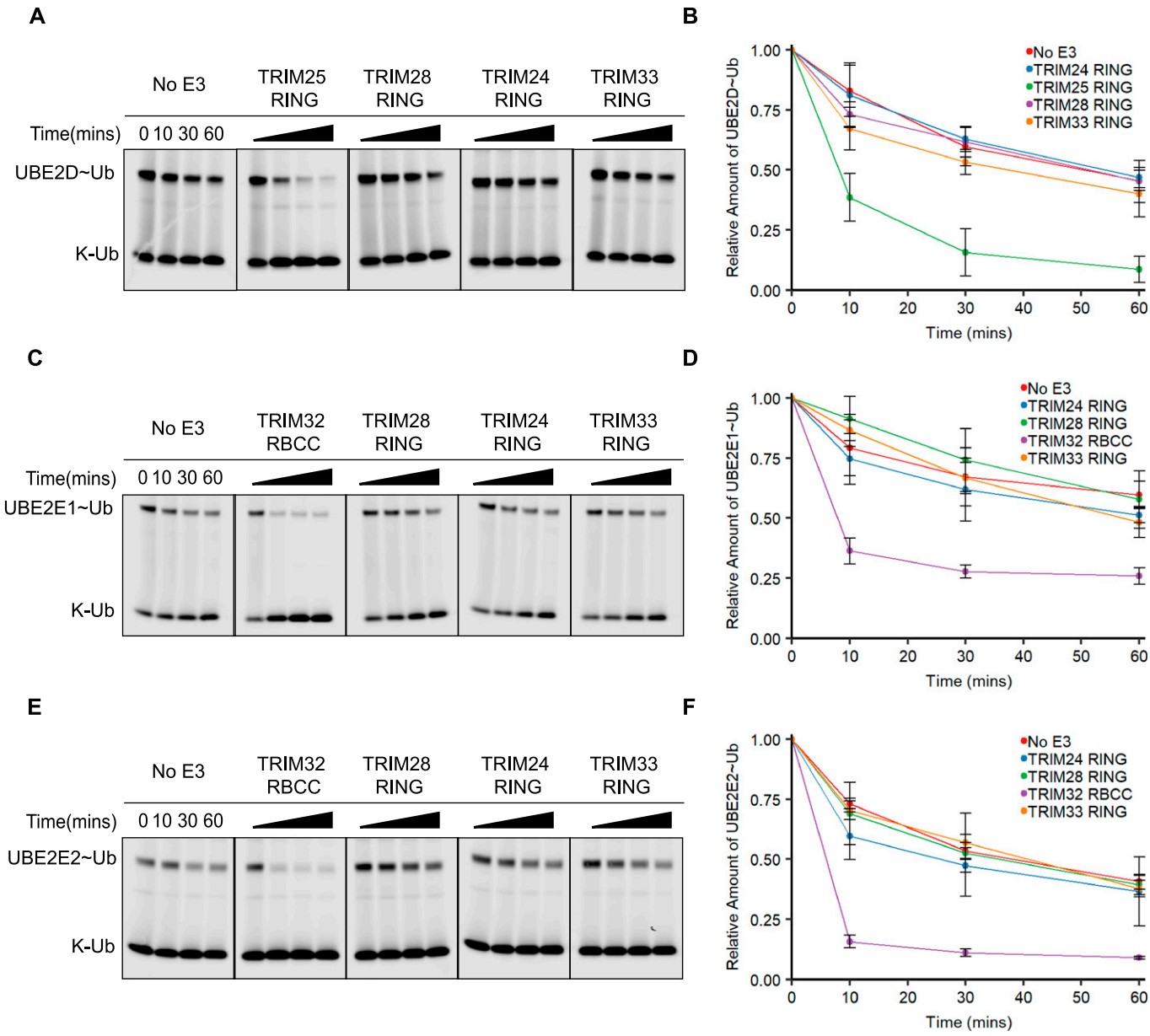

**Figure 2. Isolated class VI RING domains lack catalytic activity.**
**(A, C, E)** E2–Ub$^{Atto}$ discharge assays with class VI RING domains and different E2 enzymes as indicated. Assays were carried out with TRIMs as indicated, and reactions were monitored over 60 min. **(B, D, F)** Quantification of the discharge assays: the loss of E2–Ub$^{Atto}$ is plotted as the average of experimental triplicates or duplicates (±SD). The discharge assays were run in parallel to those shown in Figs 3D and S2. Some of the panels that act as negative and positive controls are shown in two figures.
Source data are available for this figure.

(57.5–62.2 ppm) and C$\beta$ (29.9–32.2 ppm) resonances are compatible with these residues being in a reduced form. Zn-coordinating residues were identified on the basis of sequence alignment with other TRIM RING structures (Fig 1).

$^{15}$N-NMR relaxation parameters (T$_1$ and T$_2$) and heteronuclear NOEs were obtained from the TRIM28 RING domain at 700 MHz ($^1$H) (Fig 4A). The pattern of the relaxation data reveals that the N- (aa 54–59) and C-terminal (aa 139–145) regions of the RING domain, together with the long L2 loop (aa 98–112), are highly dynamic and experience fast motion within the pico- to nanosecond timescale.

In most dimeric RING proteins, the regions adjacent to the core RING form helices that mediate dimerisation. However, in the case of TRIM28, these regions appear flexible with backbone atomic chemical shifts typical of a random coil. The overall rotational isotropic correlation time, estimated from the T$_1$ and T$_2$ values across the structured segments of TRIM28 (aa 60–97 and 113–137), is 6.7 ± 0.1 ns. This value is similar to that of the monomeric form of the TRIM25 RING (6.4 ± 0.1 ns) (39), confirming that the TRIM28 RING is monomeric in solution at the concentration used in NMR experiments (1 mM).

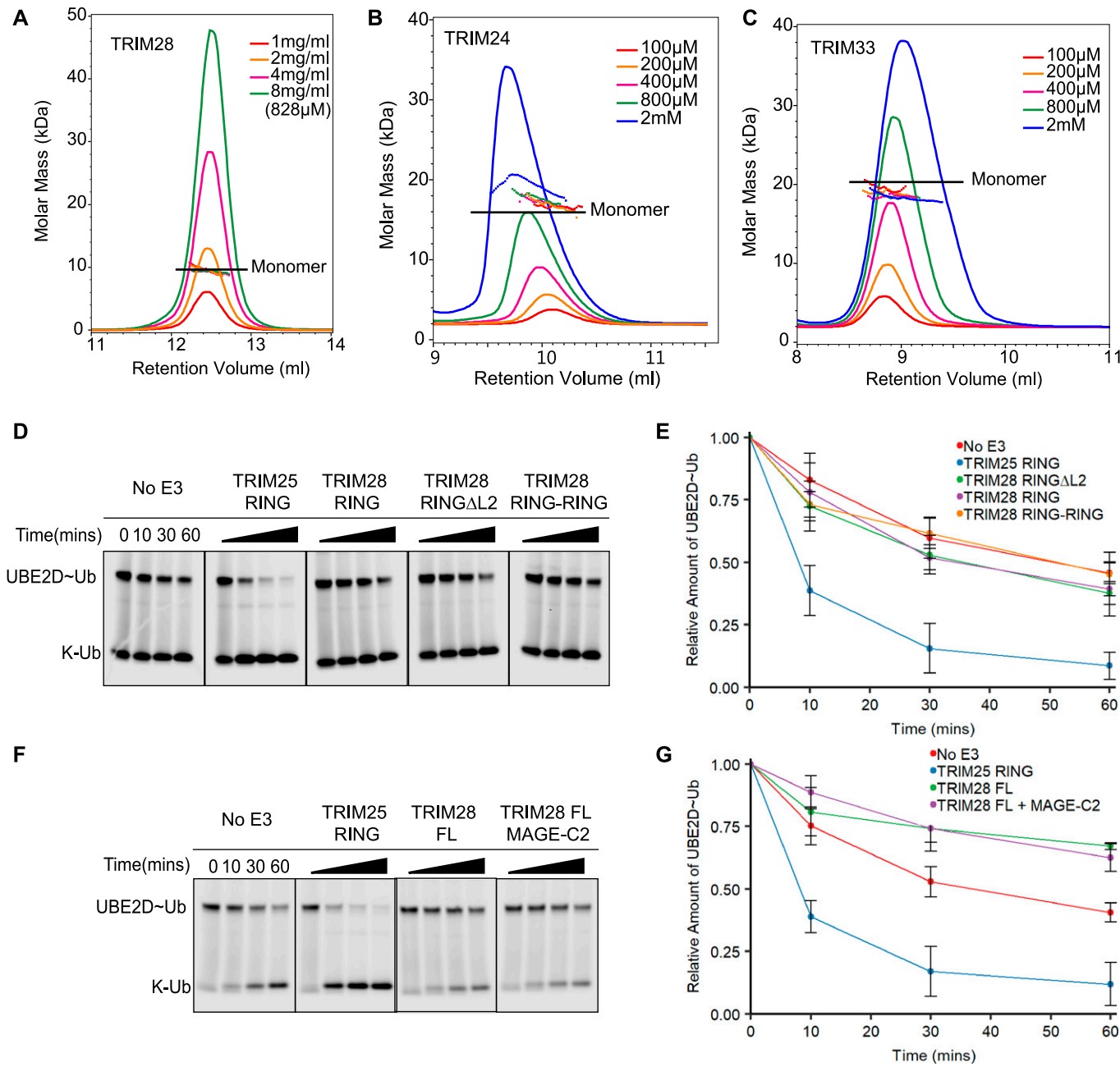

**Figure 3. Oligomeric state of class VI RINGs and link to catalytic activity.**
**(A–C)** SEC-MALLS traces of TRIM28, TRIM24, and TRIM33. The constructs were analysed over different concentration ranges, as indicated. **(D, F)** UBE2D–Ub[Atto] discharge assays with various TRIM28 constructs ± MAGE as indicated. Reactions were monitored over 60 min. **(E, G)** Quantification of the discharge assays: the loss of UBE2D–Ub[Atto] is plotted as the average of experimental triplicates (±SD). The discharge assays were run in parallel to those shown in Figs 2A, C, E, and S2. Some of the panels that act as negative and positive controls are shown in two figures.
Source data are available for this figure.

Three-dimensional [15]N- and [13]C-separated [1]H-NOESY were then used to determine the solution structure of the TRIM28 RING. The representative lowest energy structure and the superimposed 20 lowest energy conformers are shown in Fig 4B. Initial structural conformers were calculated in the absence of zinc ions to confirm the predicted pattern of coordinating residues. Final structures were calculated with a number of zinc restraints to ensure the correct coordination geometry around the ions. A total of 1,516 NMR data-derived distance restraints and 108 $\varphi/\psi$ backbone angle constraints were used to calculate the final structure. TRIM28 adopts a classical RING domain fold, characterised by a cross-brace pattern of cysteine and histidine residues coordinating two $Zn^{2+}$ ions, with no violations of the distance or angle experimental restraints observed above 0.5 Å or 5°, respectively. As predicted, the highly dynamic L2 loop is not well defined in the final 20 lowest energy conformers, indicating that it is unlikely to be engaged in intramolecular, autoinhibitory interactions.

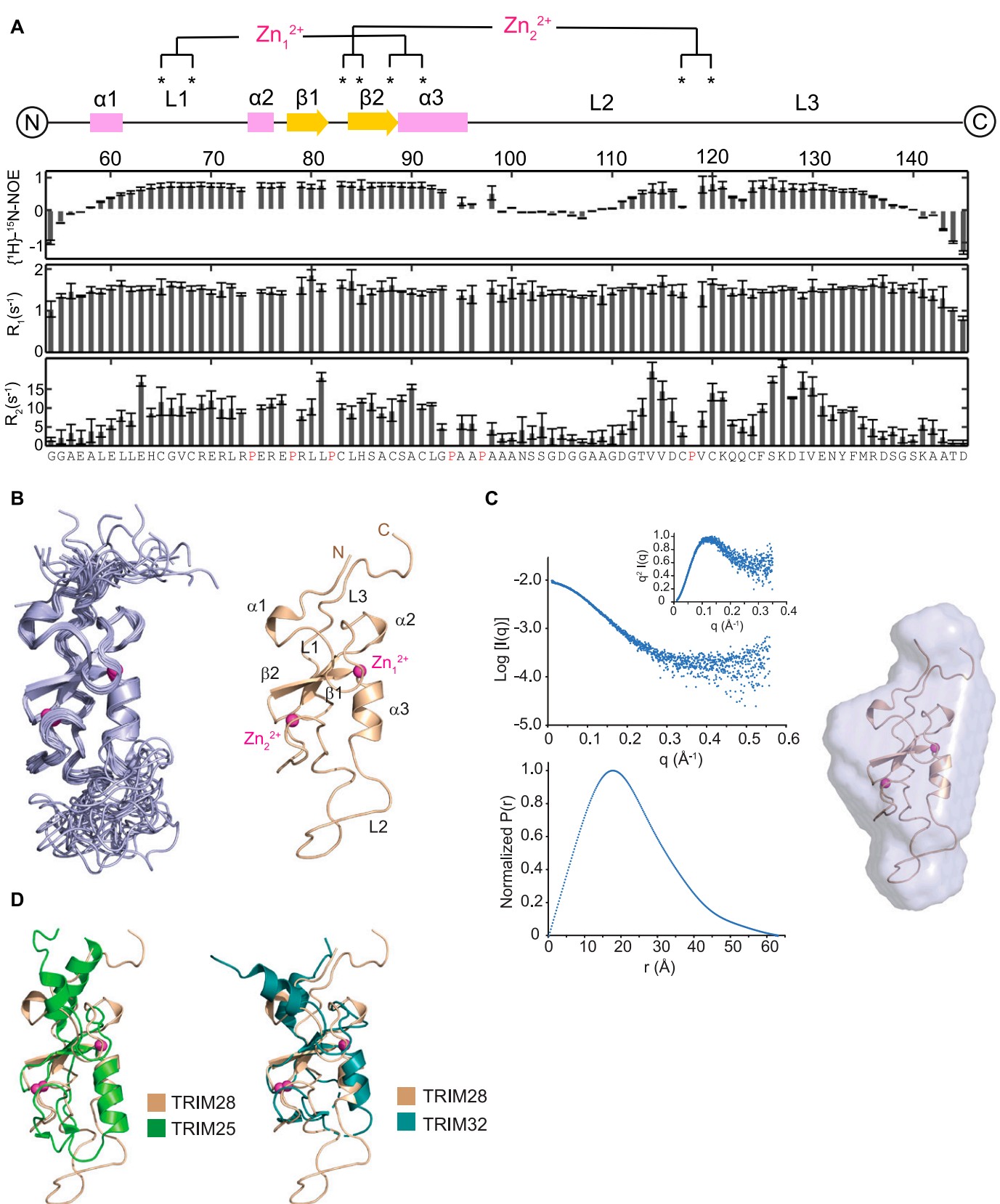

The C terminus of the protein is largely unstructured, whereas there is a small helical turn (α1) at the N terminus that is reminiscent of the extended helical region in dimeric TRIMs.

To validate the RING solution structure, we carried out small angle X-ray scattering (SAXS) analysis of the same construct. X-Ray scattering curves of the TRIM28 RING domain were acquired during fractionation of the protein, at millimolar concentration, by analytical gel filtration. The averaged scattering profile of the elution peak is reported in Fig 4C, together with the normalized pair-distribution function and the Kratky plot. The SAXS parameters and data statistics are reported in Table 1. The P(r) distribution function is that of a slightly prolate protein with a $D_{max}$ of 61.0 Å and $R_g$ of 17.4 Å. The SAXS-derived molecular mass is in agreement with the calculated RING mass, whereas the Kratky plot shows a protein with a large degree of conformational flexibility. The lowest energy ab-initio model protein envelope obtained by the program DAMMIF is shown in Fig 4C, overlapped to the lowest energy representative conformer calculated by NMR ($\chi^2$ = 3.1). The agreement between the SAXS and NMR data further validates the restraints-derived monomeric TRIM28 RING structure.

Together, these data indicate that the core RING domain fold of TRIM28 is similar to that observed in other RING structures (Fig 4D), whereas dimerisation is likely impeded by the absence of extended α-helices at the N and C termini.

### TRIM28 interaction with E2–Ub conjugate

Although our analysis of the TRIM28 RING domain by MALLS, NMR and SAXS confirms that this protein is monomeric in isolation, it is possible that the RING may dimerise upon interaction with a charged E2–Ub conjugate. To test this possibility, and establish its general ability to interact with E2 and ubiquitin, we carried out NMR titration series with $^{15}$N-labelled TRIM28 RING and either UBE2D, UBE2D-Ub, or ubiquitin. We monitored chemical shift perturbation in the 2D $^{1}$H-$^{15}$N HSQC spectrum of the RING, produced by either self-association or interactions with UBE2D or ubiquitin.

The addition of four equivalents of UBE2D did not produce any significant changes in the spectrum of the TRIM28 RING (Fig S5B). In contrast, addition of four equivalents of UBE2D–Ub induced a number of chemical shift changes in the spectrum of the RING domain (Figs 5A and S5C). Mapping these shift changes to the surface of the TRIM28 RING reveals two defined regions: a larger interface, which roughly encompasses the canonical proximal RING ubiquitin-binding site, and a smaller cluster of residues in the region of the distal RING ubiquitin-binding site (Fig 5B). Interestingly, titration of ubiquitin alone also induced a number of resonance shifts in the spectrum of the TRIM28 RING, which clustered in a similar fashion to those induced by titration of the UBE2D-Ub conjugate (Figs 5C and D, and S5D). Given that titration of UBE2D alone did not induce any chemical shift perturbations and titration with ubiquitin induced the same changes as UBE2D–Ub, this suggests that all contacts

**Table 1. SAXS data collection parameters and statistics.**

| Instrument | SEC-SAXS at SWING beamline SOLEIL |
|---|---|
| SEC column | Bio-SEC 3 Agilent 100 Å |
| q range | 0.0082–0.559 Å$^{-1}$ |
| Temperature | 15°C |
| Concentration | 10 mg/ml |
| I(0)$_{reciprocal}$/I(0)$_{real}$ | 0.0089 ± 0.00002/0.0089 ± 0.00002 cm$^{-1}$/abs |
| Rg$_{reciprocal}$/Rg$_{real}$ | 17.35 ± 0.06/17.37 ± 0.08 Å |
| D$_{max}$ | 62.5 Å |
| Porod volume | 14,113 Å$^3$ |
| Mass (Vp/1.6) | 8.8 kD |
| Mass SAXSMow (q$_{max}$ = 0.23 A$^{-1}$) | 9.5 kD |
| Mass from primary sequence | 9.7 kD |
| Normalized spatial discrepancy | 0.55 ± 0.02 |
| Best DAMMIF model $\chi^2$ | 1.31 |

observed occur between TRIM28 and ubiquitin and that TRIM28 does not make any contacts with the E2.

Superposition of the TRIM28 NMR structure with the crystal structures of the TRIM25 RING dimer in complex with UBE2D1–Ub (5FER) and the TRIM32 RING dimer (5FEY) (39) revealed that a hydrophobic residue, L17, involved in E2 binding in TRIM25, and conserved in TRIM32, corresponds to an arginine residue in TRIM28. This alignment also indicated that this arginine residue would be in close proximity to an arginine residue in UBE2D1 (R5).

To determine the effect of this residue on E2 binding and activity, an arginine residue was introduced into both the TRIM25 and TRIM32 RINGs, whereas a leucine residue was introduced into the TRIM28 RING. 1D $^{1}$H-NMR analysis of the RING mutants confirmed that these point mutations did not affect folding of the respective RINGs (Fig S3B and C), and activity of the mutant proteins was analysed using lysine-discharge assays. The TRIM28 R69L mutation had no effect on activity (Fig 6A and B), whereas both the TRIM25 L17R mutation and the TRIM32 M24R mutation significantly reduced the activity of the respective RING domains (Fig 6C–F). Based on these data, we conclude that this hydrophobic residue is likely important for E2 binding in TRIM25 and TRIM32 but is not sufficient to rescue the activity of TRIM28.

## Discussion

Members of the TRIM family of E3 ligases regulate a multitude of cellular functions ranging from developmental processes to innate

**Figure 4. Structural characterisation of the TRIM28 RING domain.**
**(A)** Residue-specific $^{15}$N NMR relaxation parameters obtained for TRIM28 RING. **(B)** Left panel shows the superimposition of the NMR-derived 20 lowest energy conformers of TRIM28 RING. Right panel shows the lowest energy conformer with structural features and zinc ions indicated. **(C)** The solvent-subtracted SAXS profile, Kratky plot and P(r) distribution for TRIM28 RING (left panel), and the SAXS-derived envelope superimposed to the lowest energy conformer calculated from NMR data (right panel). **(D)** Superimposition of the TRIM28 RING NMR structure with the crystal structure of the TRIM25 RING (5FER) (left panel) and the TRIM32 RING (5FEY) (right panel).

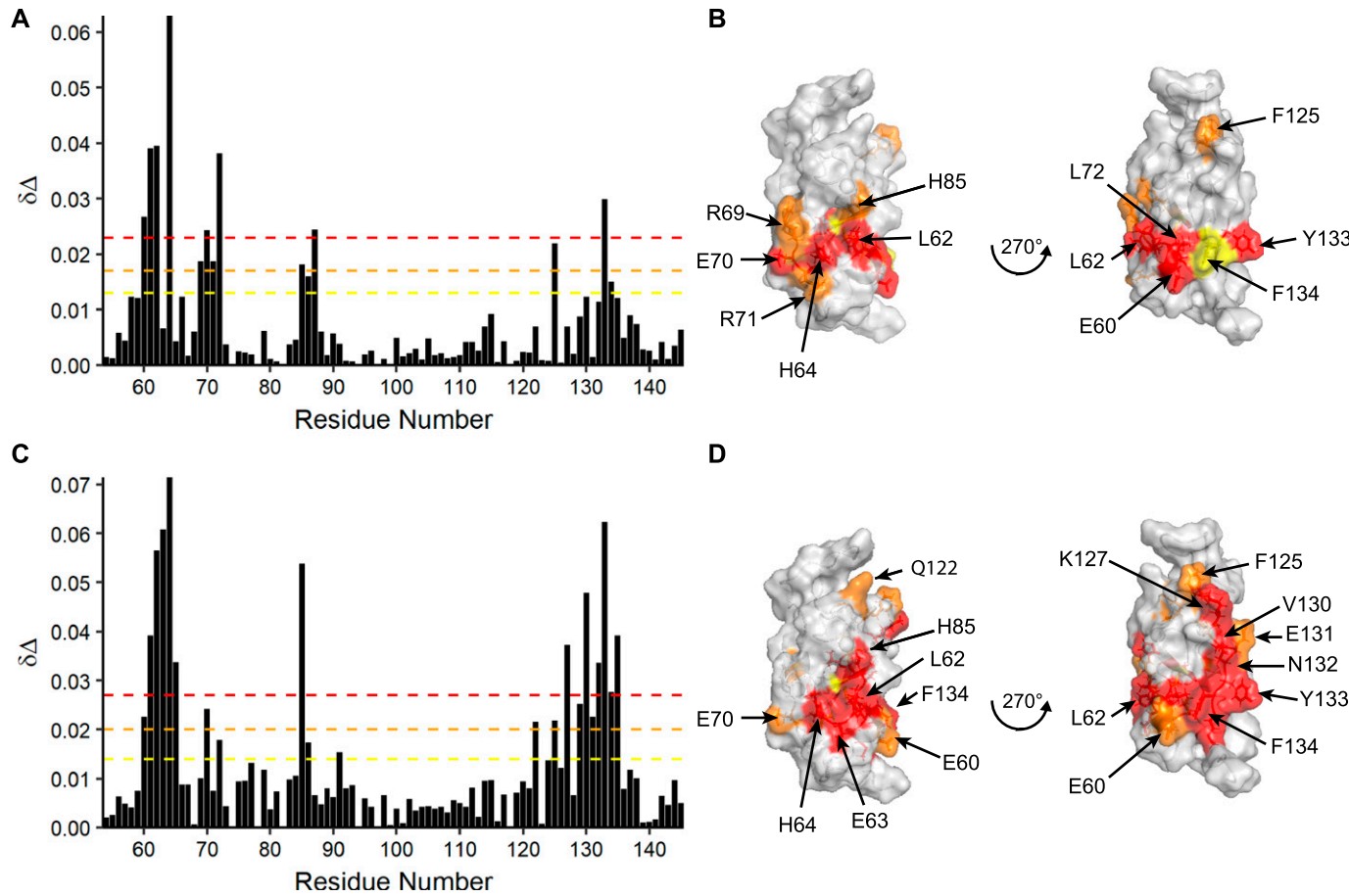

**Figure 5. Interaction of TRIM28 RING with E2–ubiquitin.**
**(A, C)** $^1$H-$^{15}$N chemical shift perturbations ($\delta\Delta$) of $^{15}$N-TRIM28 RING residues upon titration with unlabelled UBE2D–Ub (A) or ubiquitin (C) plotted against the residue number. **(B, D)** Residues that experience chemical shift changes upon titration of either UBE2D (B) or ubiquitin (D) are mapped onto the surface of TRIM28 RING. Residues are coloured to represent the relative change in chemical shift and denoted by the coloured horizontal lines on the plots (A, C).

immune signalling, and their dysregulation has been linked to many diseases. Most TRIM proteins studied thus far derive their biological function from their E3 ubiquitin ligase activity; ubiquitination may induce activation of signalling cascades, mediate protein–protein interactions, or target a protein for degradation by the proteasome. This activity appears to be largely regulated by availability of the substrate and only a few examples have been described where additional mechanisms, such as phosphorylation, modulate catalytic activity (8).

Class VI TRIM proteins are an unusual class of TRIMs as they are best known for their role as transcriptional regulators, a role which is functionally independent of their ubiquitination activity. Given this dual functionality, we hypothesised that the ubiquitination activity of class VI family members would require tight regulation to prevent degradation of components of the chromatin-remodelling complexes they form part of. To address this question, we first characterised the activity of their isolated RING domains, which we reasoned should be unaffected by any potential (auto)inhibitory mechanisms and should reflect their basic catalytic activity. To our surprise, the RING domains of TRIM24, TRIM28, and TRIM33 were unable to promote ubiquitin discharge from the E2–Ub conjugates of UBE2D, UBE2E1, and UBE2E2, E2 conjugating enzymes that were

described to work in conjunction with these TRIMs. Similarly, no activity was detected for full-length TRIM28, excluding the possibility that non-RING elements are required for E3 function.

RING dimerisation is a prerequisite for E3 ligase activity in most TRIM proteins studied so far. Dimerisation stabilises the E2–Ub conjugate in a closed conformation by allowing both RING domains to contact ubiquitin simultaneously, priming the E2–Ub thioester bond for catalysis. Our analysis of the oligomerisation properties of the isolated RINGs of TRIM24, TRIM28, and TRIM33 revealed that these proteins are constitutively monomeric in isolation. Furthermore, dimerisation could not be promoted by interaction with an E2–Ub conjugate nor by fusing two RING domains via a short linker, both of which can enhance dimerisation and catalytic activity in other RINGs that have a low propensity for self-association.

Nonetheless, not all RING-type E3s require dimerisation for activity. For example, a phosphotyrosine residue in the RING-type E3 ligase Cbl stabilises the E2–Ub conjugate in the absence of RING dimerisation, and binding of a non-covalent ubiquitin molecule to the RING backside enhances the activity of the monomeric E3 Arkadia (47, 48). Furthermore, TRIM21 was recently reported to be active as a monomer, although this RING crystallised as a dimer, indicating that a low propensity for dimerisation is maintained (8).

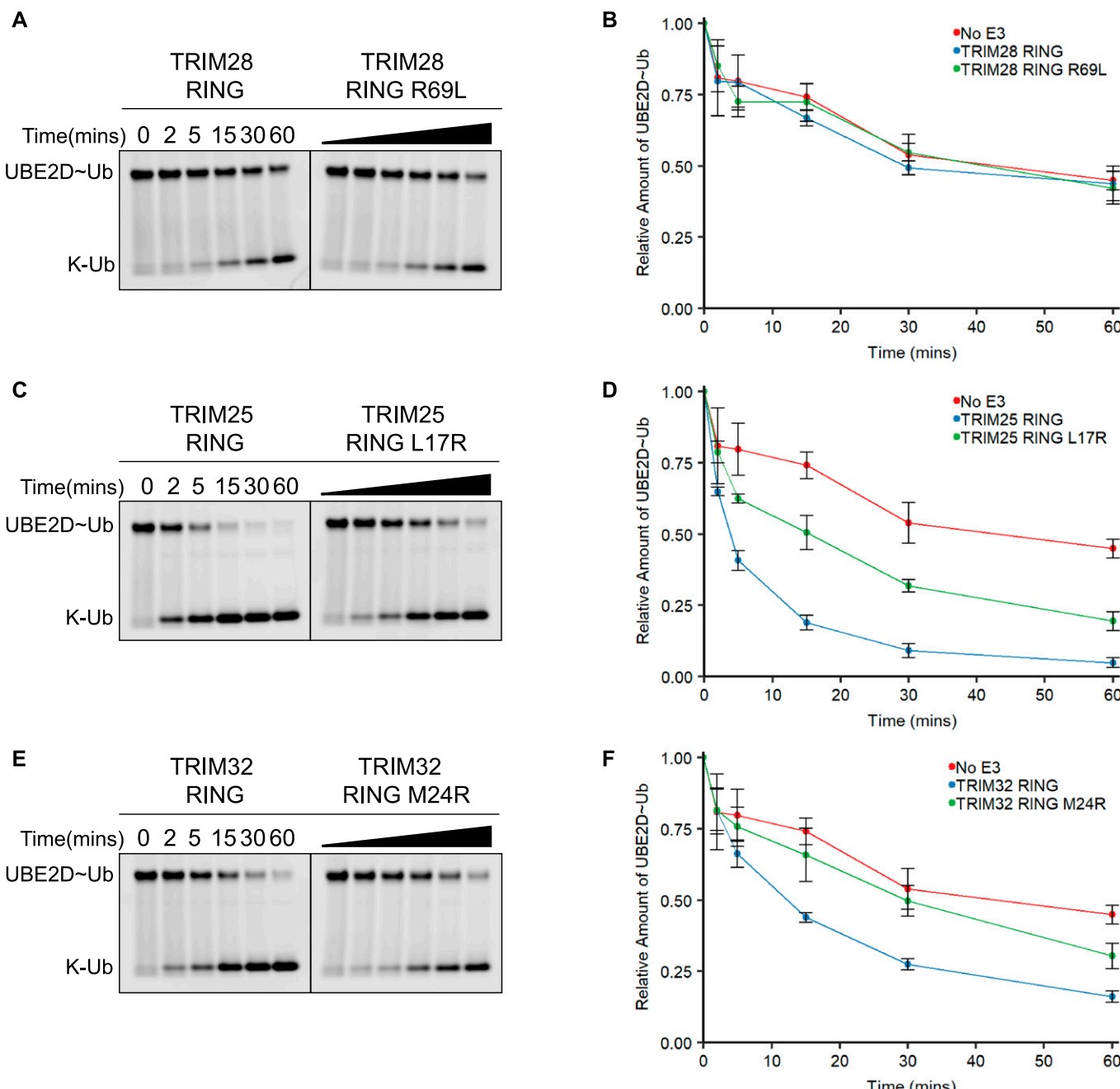

**Figure 6. Mutational analysis of RING domain catalytic activity.**
**(A, C, E)** UBE2D–Ub^Atto discharge assays with TRIM28, TRIM25, and TRIM32 RING mutants. Reactions were monitored over 60 min. **(B, D, F)** Quantification of the discharge assays: the loss of UBE2D–Ub^Atto is plotted as the average of experimental triplicates (±SD).
Source data are available for this figure.

Given that there is some precedent for active monomeric RING domains, we speculated that there would be additional structural features within the class VI RINGs, which preclude ligase activity. To identify these structural elements and determine the structural basis for the observed lack of dimerisation, we solved the solution structure of the TRIM28 RING domain. The structure reveals that TRIM28 shares the canonical RING domain fold but lacks the N- and C-terminal α-helices characteristic of dimeric TRIM RING domains. Instead, NMR relaxation analysis and heteronuclear NOEs indicate that the TRIM28

RING N and C termini are largely unstructured and highly dynamic, precluding dimerisation in the same fashion as other TRIM proteins.

Further comparison of the NMR structure of the TRIM28 RING with crystal structures of RING/E2–Ub complexes, such as that of TRIM32 or TRIM25, revealed that a conserved hydrophobic residue (L17 in TRIM25), present at the interface between the RING domain and the E2, is replaced by an arginine residue in TRIM28. Substitution of L17 in TRIM25, or the corresponding M24 in TRIM32, by arginine resulted in a significant reduction in activity of these proteins. Similarly, this

hydrophobic residue has been shown to be important for BRCA1 activity and CCR4-NOT binding to UBE2D2 (49,50).

Given that TRIM28 lacks a conserved residue known to be important for E2 binding and introduction of the TRIM28 arginine residue reduced the activity of TRIM25 and TRIM32, we investigated E3/E2–Ub complex formation by NMR to determine whether the lack of RING activity may be related to an inability to bind E2. We followed chemical shift changes in the spectrum of the TRIM28 RING domain upon addition of E2, E2–Ub conjugates, and ubiquitin. These experiments did not detect any RING-E2 interaction but indicated weak binding of ubiquitin, indicating that the arginine residue we identified may indeed preclude E2 binding. Interestingly, mutation of the TRIM28 arginine residue to the leucine residue present in TRIM25 (R69L) could not rescue activity. We, therefore, propose that both lack of RING dimerisation and lack of E2 binding are responsible for the absence of observable catalytic activity of TRIM28.

Nevertheless, a number of reports have linked TRIM24, TRIM28, and TRIM33 to ubiquitination of multiple targets, including p53. However, as many of these observations were made in cell-based assays, this activity could be explained by the presence of post-translational modifications, or cellular binding partners, which are able to promote catalytic activity. This suggests the possibility that class VI TRIM proteins constitute inactive RINGs that require hetero-dimerisation with active RING partners for physiological activity. The RING1b/Bmi1 and the BRCA1/BARD1 heterodimers are both examples of heterologous RING dimers between an "inactive" and an "active" RING domain, where one component, Bmi1 and BARD1, is unable to bind E2 (49,51,52). In this context, it is interesting that the TRIM28 RING retains some ability to bind ubiquitin, and we propose that TRIM28 could stabilise the ubiquitin molecule of the E2–Ub conjugate bound by its "active" RING partner. Alternatively, the observed activity of class VI TRIMs in a cellular context could be explained by post-translational modifications which are able to promote RING activity, as described for TRIM21 (8). However, no functionally relevant modifications have been reported for the regions surrounding the TRIM28 RING. Finally, association with specific binding partners may promote interaction with the E2–Ub conjugate and thereby stimulate activity, as has been shown for RNF168 (53, 54, 55).

As class VI TRIMs have well-defined roles in transcriptional regulation, it is not surprising that some form of regulation of E3 ligase activity exists to delineate these two functions and prevent potentially deleterious ubiquitination of proteins in TRIM protein-containing transcriptional complexes. We speculate that E3 ligase activity is supressed while these TRIM members are functioning as transcriptional regulators and that E3 ligase activity is only unleashed when protein turnover is required. This study identifies a unified profile of RING activity across a specific class of TRIM proteins and provides a first link between RING domain activity and C-terminal identity.

# Materials and Methods

## Protein production and purification

TRIM28 RING (aa 54–145), RINGΔL2, and R69L mutant, and MAGE-C2 MHD (aa 132–351) constructs were cloned into pET49b to produce GST–His$_6$ fusion proteins. The TRIM33 RING (aa 1–208) and TRIM24

RING (aa 1–153) were cloned into a modified pET49b vector to produce thioredoxin–His$_6$ fusion proteins. The TRIM28 RING–RING construct was produced by connecting two copies of TRIM28 RING (aa 54–145) by a short linker sequence GGSGSG as previously described (39) and cloned into a modified pET49b vector to produce a His$_6$-tagged protein. All plasmids were verified by DNA sequencing.

His$_6$-tagged TRIM28 FL in pFastBac was kindly provided by Peter Cherepanov, and the resulting bacmid was used to express the protein in SF21 cells, for 96 h at 27°C. Full-length TRIM28 was purified by affinity chromatography, the His$_6$-tag was removed by cleavage with TEV protease, and the protein was further purified by size-exclusion chromatography (SEC) (SEC buffer: 25 mM Hepes, pH 7.5, 150 mM NaCl, and 0.5 mM TCEP).

All other proteins were expressed in BL21(DE3) *Escherichia coli* cells in LB media supplemented with 200 µM ZnCl$_2$ at 37°C for 4 h. Proteins were purified by affinity chromatography, and tags were removed by treatment with HRV 3C protease. TRIM28 RING–RING protein was further purified by ion-exchange chromatography, and all proteins were subjected to SEC. The TRIM25 and TRIM32 RING constructs and mutants, and the TRIM32 RBCC were prepared as previously described (39).

Isopeptide-linked UBE2D1–Ub used in NMR titration studies was prepared as previously described (39, 42). His$_6$-M1C-ubiquitin was labelled with Atto 647N maleimide (Sigma-Aldrich) as described (39).

Protein concentrations were determined by UV absorption at 280 nm using calculated extinction coefficients or using the BCA assay (Thermo Fisher Scientific) for those proteins lacking absorbance at 280 nm.

## In vitro ubiquitin discharge assays

E2 enzymes were charged with ubiquitin under the following conditions: 1 µM E1, 1 µM Ubiquitin$^{Atto}$, 3 mM ATP, 4 µM E2 (reaction buffer: 50 mM Hepes, pH 7.5, 150 mM NaCl, 20 mM MgCl$_2$, and 5 mM CaCl$_2$). The reactions were incubated at 25°C for 20 min, followed by a 5-min incubation with 1U of apyrase (Sigma-Aldrich). Discharge was initiated by addition of L-lysine, at a final concentration of 20 mM, and E3, at a final concentration of 4 µM. Discharge assays were incubated at 25°C for up to 60 min, the samples were quenched by addition of SDS sample buffer and flash-freezing on dry ice at the described time intervals, and resolved by SDS–PAGE. For quantification, the gels were scanned with a LICOR CLx scanner, and the bands for E2–Ub$^{Atto}$ were integrated using the ImageStudio software package (LI-COR). Experiments were performed in duplicate (UBE2E2) or triplicate (UBE2D and UBE2E1). The scans were converted to greyscale.

## Auto-ubiquitination assays

Full-length TRIM28 or TRIM25 RBCC, at a final concentration of 4 µM, was incubated with 0.5 µM E1, 2.5 µM E2, 50 µM ubiquitin, 1 µM ubiquitin$^{Atto}$, and 10 mM ATP in a total volume of 30 µl (reaction buffer: 50 mM Hepes, pH 7.5, 150 mM NaCl, 20 mM MgCl$_2$, and 5 mM CaCl$_2$). Reactions were incubated at 30°C for 60 min, the samples were quenched by addition of SDS sample buffer and incubation at 95°C. The gels were scanned with a LI-COR CLx scanner and converted to greyscale.

**SEC-MALLS**

Analytical SEC-MALLS profiles were recorded at 16 angles using a DAWN-HELEOS-II laser photometer (Wyatt Technology) and differential refractometer (Optilab TrEX) equipped with a Peltier temperature-regulated flow cell maintained at 25°C (Wyatt Technology). 100 $\mu$l samples of purified proteins at multiple concentrations were applied to a Superdex 75 10/300 GL column (GE Healthcare) equilibrated with 25 mM Hepes, pH 7.5, 150 mM NaCl, 0.5 mM TCEP, and 3 mM NaN$_3$ at a flow-rate of 1 ml/min. The data were analysed using ASTRA 6.1 (Wyatt Technology).

**Nuclear magnetic resonance (NMR)**

$^{15}$N and $^{15}$N/$^{13}$C isotopically labelled samples were prepared by growing bacteria in M9 minimal medium using $^{15}$N-ammonium chloride and $^{13}$C$_6$-glucose as the sole nitrogen and carbon sources, respectively.

TRIM28 RING (aa 54–145) NMR spectra were recorded in NMR buffer (20 mM MES, pH 6.2, 100 mM NaCl, and 0.5 mM TCEP) at 298K on Bruker AVANCE spectrometers operating at nominal $^1$H frequencies of 700, 800, and 950 MHz. Data were acquired with Topspin (version 3.5, Bruker), processed with NMRPipe/NMRDRAW (56) and analysed with CCPN software (57).

Through–bond sequential backbone resonance assignment was obtained by combining the analysis from standard triple resonance experiments. Side-chain assignment was obtained using a combination of $^1$H, $^{15}$N TOCSY-HSQC, $^1$H, $^{13}$C-HC(C)H-TOCSY, $^1$H, $^{13}$C-HC(C)H-COSY, and aromatic $^1$H, $^{13}$C-HC(C)H-TOCSY.

$^{15}$N longitudinal (R$_1$) and transverse (R$_2$) relaxation rates and $^1$H-$^{15}$N heteronuclear NOEs were measured at 298K as previously described (58). R$_1$ and R$_2$ values were determined for each residue by fitting an exponential decay to the peak intensity. R$_1$ longitudinal recovery delays were set to 10, 100, 200, 400, 600, 800, 1,200, and 1,500 ms. R$_2$ transverse recovery delays were set to 8, 16, 24, 40, 56, 80, 104, and 128 ms. In each case, the error was determined from the fit according to the procedure implemented in CCPN analysis (57). Heteronuclear NOEs were calculated from the ratio of the cross peak intensity in the saturated versus the equilibrium spectra ($\eta = I_{sat}/I_0$) (59). The residues were excluded where overlap in the data precluded accurate measurement of the peak intensity. Isotropic correlation times were determined using the program TENSOR2 (60).

**Structure calculations**

Inter-proton distance restraints for the TRIM28 RING domain were derived from the analysis of the 3D $^1$H-$^{15}$N NOESY-HSQC, $^1$H-$^{13}$C-aromatic NOESY-HSQC, and $^1$H-$^{13}$C NOESY-HSQC spectra, all acquired with 120 ms mixing time. A large proportion of the NOESY cross peaks were initially assigned in a manual fashion, without ambiguity; the remaining cross peaks were assigned automatically through the iterative calculation scheme implemented in ARIA v2.2 (61). Backbone torsion angle restraints ($\varphi/\psi$), derived from analysis of $^1$H$\alpha$, $^{13}$C$\alpha$, $^{13}$C$\beta$, $^{13}$C′, $^{15}$N, and $^1$HN chemical shift databases by the program TALOS (62), were also used in the calculations. Both the dihedral restraints and ARIA-derived ambiguous and unambiguous distance restraints lists were then cross-checked and further improved in an iterative manner by monitoring violations during the initial stages of the structure

calculations using an Xplor-NIH–simulated annealing protocol (63). The protocol adopts a mixture of Cartesian molecular dynamics and torsion angle dynamics-simulated annealing to refine structures from random-coil starting conformers with good local geometry. A final step of restrained molecular dynamics with knowledge-based energy terms, such as torsion angle potential derived from conformational databases (64), or backbone hydrogen bond potential (65), was used to further improve the quality of the final conformers. Sequence analysis highlighted the pattern of residues in the TRIM28 RING domain that coordinate the zinc ions. Structure calculations where zinc ions were explicitly removed produced the same TRIM28 RING domain fold.

A total of 1,176 unambiguous and 319 ambiguous NOE-derived inter-proton distance restraints and 108 dihedral angle restraints were used in the final TRIM28 RING domain structure calculations. 21 distance restraints were used to ensure good geometry in the coordination of the zinc ions. The quality of the final 20 conformers was evaluated by the program PROCHECK (66). The NMR structural statistics of the 20 lowest energy structures are reported in Table 2. Coordinates were deposited in the Protein Data Bank under accession code 6I9H. Chemical shifts and restraints have been deposited in the Biological Magnetic Resonance Data Bank under accession number 34330.

**NMR titrations**

The titrations of the $^{15}$N-labelled TRIM28 RING domain with unlabelled UBE2D1, UBE2D1-ubiquitin, and ubiquitin were performed at constant concentration of the TRIM28 RING (200 $\mu$M) and TRIM28:ligand molar ratios ranging from 1:0 to 1:4, as previously described (67). Chemical shift changes for the backbone amide $^1$H and $^{15}$N nuclei ($\Delta\delta$) were calculated according to the procedure implemented in CCPN (57). Any changes in the spectrum of the labelled component during the titration can be attributed directly to an intermolecular interaction, as both proteins were pre-exchanged into the same stock buffer.

**Small-angle X-ray scattering**

SAXS data were collected at the SWING beamline at SOLEIL. The purified TRIM28 RING construct, at 10 mg/ml (1 mM), was injected onto a Bio SEC-3 100 Å Agilent column and eluted at a flow rate of 0.2 ml/min at 15°C. Frames were collected continuously during the fractionation of the proteins. Frames collected before the void volume were averaged and subtracted from the signal of the elution profile to account for background scattering. Data reduction, subtraction, and averaging were performed using the software FOXTROT (SOLEIL). The scattering curves were analysed using the package ATSAS (68). Low-resolution three-dimensional *ab initio* models for the TRIM28 RING molecular envelope were generated by the program DAMMIF (69) and overlapped to the NMR-derived structure using SUPCOMB (70). The SAXS-derived dummy atom models were rendered with the PyMOL molecular graphics system (Schrödinger, LLC).

# Supplementary Information

**Table 2. NMR-derived restraints and structural statistics of the ensemble.**

| | Ensemble (n = 20) | Lowest energy |
|---|---|---|
| Deviation from Experimental Data | | |
| All (1,516) (Å) | 0.070 ± 0.005 | 0.062 |
| Intra-residue (i = j) (589) (Å) | 0.053 ± 0.002 | 0.051 |
| Sequential (\|i–j\| = 1) (274) (Å) | 0.087 ± 0.002 | 0.089 |
| Short (1 < \|i–j\|< 5) (143) (Å) | 0.081 ± 0.008 | 0.079 |
| Long (\|i–j\|≥ 5) (170) (Å) | 0.065 ± 0.008 | 0.080 |
| Ambiguous (319) (Å) | 0.066 ± 0.005 | 0.062 |
| Zinc (21) (Å) | 0.076 ± 0.013 | 0.056 |
| Dihedral angle constraints (108) (°) | 0.86 ± 0.1 | 0.88 |
| Deviation from idealized covalent geometry | | |
| Bonds (1,317) (Å) | 0.004 ± 0.001 | 0.004 |
| Angles (2,384) (°) | 0.56 ± 0.02 | 0.58 |
| Improper dihedrals (667) (°) | 0.41 ± 0.01 | 0.43 |
| Restraint violations | | |
| NOE violations >0.5 Å | 0 | 0 |
| Dihedral angle violations >5° | 0 | 0 |
| Precision of the ensemble | | |
| Average pairwise RMSD (Å) | | |
| Backbone atoms (residues 59–97 and 113–137) | 0.7 ± 0.2 | |
| Heavy atoms (residues 59–97 and 113–137) | 1.3 ± 0.3 | |
| Quality scores | | |
| Ramachandran statistics (PROCHECK) | | |
| Most favoured region (%) | 69.9 ± 1.8 | 73.3 |
| Additionally allowed region (%) | 23.2 ± 2.8 | 20.0 |
| Generously allowed region (%) | 6.9 ± 1.2 | 6.7 |
| Disallowed region (%) | 0 | 0 |
| Number of bad contacts | 3.4 ± 1.5 | 2 |

# Acknowledgements

We thank Geoff Kelly and Alain Oregioni of the MRC Biomedical NMR Centre for access and advice and the staff of the SWING beamline at SOLEIL, Gif-sur-Yvette for synchrotron access. We thank Ian Taylor for help with SEC-MALLS and Peter Cherepanov for kindly donating a plasmid containing full-length TRIM28. We also thank Sven Kjaer for help with insect cell expression and Andrew Purkiss and Phil Walker (all Structural Biology Science Technology Platform, Francis Crick Institute) for computational support. This work was supported by the Francis Crick Institute, which receives its core funding from Cancer Research UK (FC001142), the UK Medical Research Council (FC001142), and the Wellcome Trust (FC001142).

## Author Contributions

RV Stevens: conceptualization, formal analysis, investigation, methodology, and writing—original draft, review, and editing.
D Esposito: conceptualization, data curation, formal analysis, investigation, methodology, and writing—original draft, review, and editing.

K Rittinger: conceptualization, supervision, funding acquisition, and writing—review and editing.

## Conflict of Interest Statement

The authors declare that they have no conflict of interest.

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
