## [Reviewer comments · Life Science Alliance]

Life Science Alliance

Characterisation of Class VI TRIM RING domains: linking RING activity to C-terminal domain identity

Rebecca Stevens, Diego Esposito, and Katrin Rittinger

DOI: <https://doi.org/10.26508/lisa.201900295>

Corresponding author(s): Katrin Rittinger, The Francis Crick Institute

Review Timeline:	Submission Date:	2019-01-07
	Editorial Decision:	2019-02-08
	Revision Received:	2019-04-01
	Editorial Decision:	2019-04-11
	Revision Received:	2019-04-12
	Accepted:	2019-04-16

Scientific Editor: Editor Life Science Alliance

Transaction Report:

February 8, 2019

Re: Life Science Alliance manuscript #LSA-2019-00295-T

Dr. Katrin Rittinger
The Francis Crick Institute
Molecular Structure of Cell Signalling Laboratory
1 Midland Road
London NW1 1AT
United Kingdom

Dear Dr. Rittinger,

Thank you for submitting your manuscript entitled "Characterisation of Class VI TRIM RING domains: linking RING activity to C-terminal domain identity" to Life Science Alliance. The manuscript was assessed by expert reviewers, whose comments are appended to this letter.

As you will see, the reviewers appreciate your data. However, they also think that some of your conclusions need better support, and they provide constructive input on how to provide this additional support. I would thus like to invite you to submit a revised version of your work, addressing the individual points raised by the reviewers.

Thank you for this interesting contribution to Life Science Alliance. We are looking forward to receiving your revised manuscript.

Sincerely,

- A letter addressing the reviewers' comments point by point.
- An editable version of the final text (.DOC or .DOCX) is needed for copyediting (no PDFs).
- High-resolution figure, supplementary figure and video files uploaded as individual files: See our detailed guidelines for preparing your production-ready images, <http://life-science-alliance.org/authorguide>
- Summary blurb (enter in submission system): A short text summarizing in a single sentence the study (max. 200 characters including spaces). This text is used in conjunction with the titles of papers, hence should be informative and complementary to the title and running title. It should describe the context and significance of the findings for a general readership; it should be written in the present tense and refer to the work in the third person. Author names should not be mentioned.

B. MANUSCRIPT ORGANIZATION AND FORMATTING:

Full guidelines are available on our Instructions for Authors page, <http://life-science-alliance.org/authorguide>

Reviewer #1 (Comments to the Authors (Required)):

In the manuscript by Stevens et al., the activity of class VI TRIM RING domains were analyzed in vitro through biochemical and biophysical approaches. Overall, this is a nice study that highlights potential differences between TRIM proteins. Most interestingly, the authors propose that the class VI TRIMs that function as epigenetic modifiers have evolved mechanisms to toggle ligase activity

on/off when needed. However, this conclusion is mostly drawn from negative data.

1) Figure 2: Given the intrinsic discharge of Ub over the course of the 60 min experiment reaches 50%, the authors should test increasing concentrations of the RINGs at an earlier time point (~15 min).

2) Figure 3: Most of the studies are done with isolated RING domains and examining ubiquitin discharge instead of auto-ubiquitination due to lack of lysines on the RINGs. However, Fig. 3F does use full-length protein and thus testing auto-ubiquitination as an alternative readout would be suggested.

3) Figure 2-3: The conclusions of the study would be further supported by testing of additional E2 enzymes. Currently only 3 E2 enzymes have been tested.

4) Figure 2-3: The lack of activity is ultimately attributed to the absence of key binding partners or post-translational modifications. The impact of the study would be substantially increased if the authors could establish this conclusion. For example, activity assays could be done from protein isolated from cell extracts (insect or mammalian) to preserve binding partners and PTMs.

Reviewer #2 (Comments to the Authors (Required)):

In this manuscript Stevens et al. characterize the ring domain of class VI TRIM proteins. Based on lysine discharge assays they showed that the RING domain of this subfamily is inactive. Then they showed using SEC MALS that these ring domains are monomeric. In parallel, they described the solution structure of TRIM28 RING domain and suggested a similar fold to the one observed in other RING structures. Based on their results, they suggest that additional proteins are required for E3 activity.

1. The authors tested E3 activity based on lysine destabilization and suggested that these proteins lack E3 activity and need additional partner/s. However, Alton et al (PNAS 2009) have shown in vitro activity for TRIM24 against P53 suggesting that additional proteins are not required (fig 4D). Therefore, it is not clear whether additional proteins are required for E3 activity, or only additional domains such as the B-box. The authors should test this possibility.

2. The authors generated a tandem ring domain of TRIM to facilitate dimerization and then showed that it does not contribute to the RING activity. However, it is unclear how the authors know that this fusion enables functional dimerization. Did the authors use a linker between the domains? Does fusion of the RING domain to GST, which forms dimer, enable activity?

3. The authors suggest that the interaction of the E2 with the RING domain is mediated only via the Ub, and free E2 does not bind the RING of TRIM28. This is a surprising observation that raises the question of whether TRIM 28 has E3 ligase activity at all. Did the authors detect interaction of TRIM25 with the E2 using NMR? Also, did they check the binding of TRIM 28 with R to L mutation to E2 using NMR? The authors suggest that the lack of E2 binding leads to no E3 ligase activity. To support this possibility the authors can show that TRIM 28 or 24 possessing the RING domain of TRIM 25 recovers an E3 ligase activity.

Reviewer #1

In the manuscript by Stevens et al., the activity of class VI TRIM RING domains were analyzed in vitro through biochemical and biophysical approaches. Overall, this is a nice study that highlights potential differences between TRIM proteins. Most interestingly, the authors propose that the class VI TRIMs that function as epigenetic modifiers have evolved mechanisms to toggle ligase activity on/off when needed. However, this conclusion is mostly drawn from negative data.

1) Figure 2: Given the intrinsic discharge of Ub over the course of the 60 min experiment reaches 50%, the authors should test increasing concentrations of the RINGs at an earlier time point (~15 min).

We have added new data in which we monitor ubiquitin discharge from E2~Ub at 15 min at increasing concentrations of E3 (4, 20, 100 μ M). These experiments show that an increase in TRIM28 concentration does not lead to an increase in ubiquitin discharge. This observation is in agreement with our MALLS data and supports our conclusion that the lack of self-association of Class VI TRIM RING domains is, at least in part, responsible for the observed lack of activity.

We have added these data as Figure S1.

2) Figure 3: Most of the studies are done with isolated RING domains and examining ubiquitin discharge instead of auto-ubiquitination due to lack of lysines on the RINGs. However, Fig. 3F does use full-length protein and thus testing auto-ubiquitination as an alternative readout would be suggested.

We agree that auto-ubiquitination of the full-length protein is a good alternative readout for ligase activity and have carried out this experiment, using TRIM25 as a positive control. Furthermore, we have included additional E2 enzymes in this experiment in response to point 3. Unfortunately, none of the E2s tested showed activity in combination with full-length TRIM28.

These new data are now shown in Figure S4.

3) Figure 2-3: The conclusions of the study would be further supported by testing of additional E2 enzymes. Currently only 3 E2 enzymes have been tested.

We have tested a number of additional E2 enzymes in auto-ubiquitination assays with full-length TRIM28.

These new data are now shown in Figure S4.

4) Figure 2-3: The lack of activity is ultimately attributed to the absence of key

binding partners or post-translational modifications. The impact of the study would be substantially increased if the authors could establish this conclusion. For example, activity assays could be done from protein isolated from cell extracts (insect or mammalian) to preserve binding partners and PTMs.

We agree that it is important to identify the factors that may promote activity of Class VI TRIM ligases. To follow up on the suggestion of this reviewer we have carried out an experiment, in which we overexpressed HA-TRIM28 in HEK293T cells, affinity-captured HA-TRIM28 and carried out an *in vitro* ubiquitination assay without protein elution from HA-beads (adding recombinant E1, UbcH5 and ubiquitin). TRIM25 was used as a positive control. While we could detect robust auto-ubiquitination of TRIM25 in this experiment, we could not detect activity of TRIM28. This suggests that association with binding partners or post-translational modifications are inducible events. While we agree that it is important to identify the nature of such events, we believe that this is beyond the scope of this manuscript.

Reviewer #2

In this manuscript Stevens et al. characterize the ring domain of class VI TRIM proteins. Based on lysine discharge assays they showed that the RING domain of this subfamily is inactive. Then they showed using SEC MALS that these ring domains are monomeric. In parallel, they described the solution structure of TRIM28 RING domain and suggested a similar fold to the one observed in other RING structures. Based on their results, they suggest that additional proteins are required for E3 activity.

1. The authors tested E3 activity based on lysine destabilization and suggested that these proteins lack E3 activity and need additional partner/s. However, Alton et al (PNAS 2009) have shown *in vitro* activity for TRIM24 against P53 suggesting that additional proteins are not required (fig 4D). Therefore, it is not clear whether additional proteins are required for E3 activity, or only additional domains such as the B-box. The authors should test this possibility.

We have tested if additional domains might be required for activity in TRIM28, by carrying out E2~Ub discharge assays, as well as auto-ubiquitination assays with the full-length protein but could not detect any activity compared to a TRIM25 positive control. In these assays we used full-length TRIM28 that was purified to homogeneity as judged by Coomassie staining.

We acknowledge that Alton et al. have shown activity of TRIM24 against *in vitro* translated ³⁵S-p53 and can only speculate why we can't detect activity for the RING domain. The TRIM24 protein used in this study was affinity-captured from insect cells but not further purified and hence may have been associated with a binding partner. This will require further studies.

We would like to stress that we are not disputing previous reports that have described E3 ligase activity for Class VI TRIM ligases. Instead, the key message from our paper is that unlike most TRIM E3 RING domains studied thus far, which are active on their own but require homodimerisation for activity, Class VI RINGs have no tendency to dimerise and show no apparent activity without additional events (binding partners, PTMs), which remain to be identified.

2. The authors generated a tandem ring domain of TRIM to facilitate dimerization and then showed that it does not contribute to the RING activity. However, it is unclear how the authors know that this fusion enables functional dimerization. Did the authors use a linker between the domains? Does fusion of the RING domain to GST, which forms dimer, enable activity?

We apologise that we have not made it clear how the fusion protein was generated. This is now described in Materials and Methods:

“The TRIM28 RING-RING construct was produced by connecting two copies of TRIM28 RING (aa 54-145) by a short linker sequence GGSGSG as previously described [39]....”

As described in reference 39 we have designed the linker connecting two RING domains based on the crystal structure of the TRIM25 RING dimer and showed biochemically that it enables functional dimerization as its activity was significantly higher than that of the isolated RING (see Ref 39).

3. The authors suggest that the interaction of the E2 with the RING domain is mediated only via the Ub, and free E2 does not bind the RING of TRIM28. This is a surprising observation that raises the question of whether TRIM 28 has E3 ligase activity at all. Did the authors detect interaction of TRIM25 with the E2 using NMR?

We did not investigate the interaction between TRIM25 and E2 using NMR as the TRIM25 RING is in a monomer-dimer equilibrium, making the interpretation of any chemical shifts observed upon addition of E2 very difficult as multiple species would be present upon complex formation.

Instead, we have done the experiment requested using the TRIM32 RING as this is a constitutive dimer. Addition of 2 and 4 equivalents of UBE2D to ¹⁵N-labelled TRIM32 shows reduction of cross peak intensities and chemical shift perturbations compatible with the formation of a higher MW complex. A small number of signals, likely derived from flexible regions and side chain amide protons, stays unperturbed indicating that the line broadening is not due to protein precipitation.

We prefer not to include these data in the manuscript as we believe it would detract from the main message.

Also, did they check the binding of TRIM 28 with R to L mutation to E2 using NMR?

We did not check binding of the TRIM28 RING R69L mutation by NMR as this mutant did not recover activity.

The authors suggest that the lack of E2 binding leads to no E3 ligase activity. To support this possibility the authors can show that TRIM 28 or 24 possessing the RING domain of TRIM 25 recovers an E3 ligase activity.

As described on page 7 of our manuscript, we had indeed tried to create TRIM chimeras where we have inserted the TRIM28 RING domain between the dimer-forming helices of TRIM32 and of TRIM2 to test if enforced dimerization may rescue activity, but neither produced a soluble protein. These experiments suggest that RING domains can't simply be exchanged between different TRIM ligases, and it is hence unclear to us what additional insight might be gained from a chimera in which the RING of TRIM25 was incorporated into a full-length Class VI TRIM.

April 11, 2019

RE: Life Science Alliance Manuscript #LSA-2019-00295-TR

Dr. Katrin Rittinger
The Francis Crick Institute
Molecular Structure of Cell Signalling Laboratory
1 Midland Road
London NW1 1AT
United Kingdom

Dear Dr. Rittinger,

Thank you for submitting your revised manuscript entitled "Characterisation of Class VI TRIM RING domains: linking RING activity to C-terminal domain identity". As you will see, the reviewers appreciate the introduced changes and we would thus be happy to publish your paper in Life Science Alliance pending final revisions necessary to meet our formatting guidelines:

- Please note that the legend to Figure 5 currently mentions a panel 'F', please correct.
- For consistency, please either call out all sub-panels for Figure S3 or none (currently panel A is called out).
- Though you provide the source data for the discharge assays, I think it would be good to mention that the experiments for Fig 2, 3 and S2 were run in parallel and that there are therefore some blots shown in several figures.

A. FINAL FILES:

- An editable version of the final text (.DOC or .DOCX) is needed for copyediting (no PDFs).
- High-resolution figure, supplementary figure and video files uploaded as individual files: See our detailed guidelines for preparing your production-ready images, <http://www.life-science-alliance.org/authors>
- Summary blurb (enter in submission system): A short text summarizing in a single sentence the

study (max. 200 characters including spaces). This text is used in conjunction with the titles of papers, hence should be informative and complementary to the title. It should describe the context and significance of the findings for a general readership; it should be written in the present tense and refer to the work in the third person. Author names should not be mentioned.

B. MANUSCRIPT ORGANIZATION AND FORMATTING:

Sincerely,

Andrea Leibfried, PhD
Executive Editor
Life Science Alliance
Meyrhofstr. 1
69117 Heidelberg, Germany
t +49 6221 8891 502
e a.leibfried@life-science-alliance.org
www.life-science-alliance.org

Reviewer #1 (Comments to the Authors (Required)):

All of my concerns have been addressed.

Reviewer #2 (Comments to the Authors (Required)):

The authors have further improved their manuscript by addressing the issues raised during review process.

April 16, 2019

RE: Life Science Alliance Manuscript #LSA-2019-00295-TRR

Dr. Katrin Rittinger
The Francis Crick Institute
Molecular Structure of Cell Signalling Laboratory
1 Midland Road
London NW1 1AT
United Kingdom

Dear Dr. Rittinger,

Thank you for submitting your Research Article entitled "Characterisation of Class VI TRIM RING domains: linking RING activity to C-terminal domain identity". It is a pleasure to let you know that your manuscript is now accepted for publication in Life Science Alliance. Congratulations on this interesting work.

DISTRIBUTION OF MATERIALS:

Again, congratulations on a very nice paper. I hope you found the review process to be constructive and are pleased with how the manuscript was handled editorially. We look forward to future exciting submissions from your lab.

Sincerely,

Andrea Leibfried, PhD
Executive Editor
Life Science Alliance